# The prevalence of under nutrition and associated factors among pregnant women attending antenatal care service in public hospitals of western Ethiopia

Lamessa Tafara[1], Haile Bikila[2]*, Ilili Feyisa[2], Markos Desalegn[2], Zalalem Kaba[3]

1 Lalo Asabi District, Enango Town, Western Ethiopia, 2 Department of Public Health, Institute of Health Sciences, Wollega University, Nekemte, Ethiopia, 3 Water, Sanitation, Hygiene (WASH), and Neglected Tropical Diseases (NTDs) Program Coordinator, Nekemte, Western Ethiopia

* haile.bikila@gmail.com

**Data Availability Statement:** All relevant data are within the paper and its Supporting information files.

## Abstract

### Background

Pregnancy is a time when the body is under a lot of stress, which increases your dietary needs. Under nutrition is a worldwide health issue, especially among pregnant women. Malnutrition during pregnancy can result in miscarriages, fetal deaths during pregnancy, preterm delivery, and maternal mortality for both the mother and her fetus. Therefore, this research aimed to assess the prevalence of under nutrition and associated factors among pregnant woman attending antenatal care services at public hospitals in west Ethiopia.

### Objective

To assess the prevalence of under nutrition and associated factors among pregnant women attending Antenatal Care service in Public Hospitals of western Ethiopia.

### Methods

Facility based cross-sectional study was conducted from April 10 to May 10, 2020 among 780 pregnant mothers. The study participants were selected by systematic random sampling methods from antenatal care clinics of the hospitals. Interviewer administered structured questionnaire was used to collect the data and Mid-upper arm circumference, height and weight were measured to determine the magnitude of under nutrition among the study participants. The data were entered to Epi Info version 7.2.3, and then exported to SPSS version 24 for analysis. Multivariable logistic regression was used to identify independent predictors considering adjusted odd ratio (AOR) at p-value $\leq$ 0.05 to measure the strength of association between dependent and independent variables.

### Result

The prevalence of under nutrition among pregnant women was found to be 39.2% (95%CI: 35.7%, 42.6%). Rural residence [(AOR = 1.97, 95% CI: (1.24, 3.14)], substance use [(AOR:

**Funding:** The author(s) received no specific funding for this work.

**Competing interests:** The authors have declared that no competing interests exist.

**Abbreviations:** ANC, Antenatal Care; AOR, Adjusted Odds Ratio; BSc, Bachelors of science; COR, Crude Odds Ratio; DALYs, Disability Adjusted Life Years; FAO, Food And Agricultural Organization; FFQ, Food Frequency Questionnaire; FHI, Family Health International; GDP, Gross Domestic Product; GHI, Global Hunger Index; HFIAS, Household Food Insecurity Access Scale; MCH, Maternal And Child Health; MDDW, Minimum Dietary Diversity Of Women; MUAC, Mid Upper Arm Circumference; SDGs, Sustainable Development Goals; UN, United Nations; UNICEF, United Nations International Children's Emergency Fund; USA, United States of America; WHO, World Health Organization.

3.33, 95% CI: (1.63, 6.81)], low dietary diversity of women [(AOR = 7.56, 95% CI: (4.96, 11.51)], mildly food insecure household [(AOR = 4.36, 95% CI: (2.36, 8.79)], moderately food insecure household [(AOR = 3.71, 95%CI: (1.54, 8.79), and severely food insecure household [(AOR = 6.96, 95% CI: (3.15, 15.42)] were factors significantly associated with under nutrition.

## Conclusion

The study showed that the prevalence of under nutrition is very high among pregnant women. Factors associated with under nutrition of pregnant women were rural residency, household food insecurity, dietary diversity and substance use. All concerned bodies should made efforts to reduce the risk of under nutrition by reducing substance use and improving household food security there by to increase women's dietary diversity.

## 1. Background

Under nutrition is the result of inadequate intake of food in terms of either quantity or quality, poor utilization of nutrients due to infections or other illnesses, or a combination of these immediate causes [1]. Pregnancy strongly depends on the health and nutritional status of women, and a high proportion of pregnant women are affected by poor nutrition, which leads them to unhealthy and distressing conditions. Under nutrition goes beyond calories and signifies deficiencies in any or all of the following: energy, protein, and/or essential vitamins and minerals [2].

Pregnancy causes significant physiological stress, which increases nutritional requirements. If these demands are inadequate, not only the nutritional status of the subject will be affected, but also the course of pregnancy and lactation. Nutrition-related problems form the core of many current issues in women's health, and poor nutrition can have profound effects on reproductive outcomes [3]. A lack of adequate nutrition of good quality and quantity during pregnancy can cause health problems for both the mother and her fetus. Under nutrition is among the most common causes of maternal mortality [1, 4].

The prevalence of undernourishment of the percentage of the population without regular access to adequate calories-has stagnated since 2015, and the number of people who are hungry has actually risen to 822 million from 785 million in 2015 [1]. Expectant and nursing mothers, infants and children constitute the most vulnerable segments of a population from the nutritional standpoint. The Global Burden of Disease Study 2013 identified that maternal and child malnutrition causes 1.7 million deaths and 176.9 million DALYs (Disability Adjusted Life Years [5]. A survey carried out in South India revealed that among poor women whose diets during pregnancy provided 1400–1500 calories and about 40 g of protein daily, nearly 20% of pregnancies had terminated in abortions, miscarriages or stillbirths [6].

Maternal under nutrition directly or indirectly causes about 3.5 million deaths of women in developing countries [7]. In developing countries, it has been estimated that poor nutritional status in pregnancy accounts for 14% of fetuses with IUGR (interauterine growth restriction), and maternal stunting account for a further 18.5% [8]. If adolescents or women are under-nourished during pregnancy, the cycle of maternal malnutrition, fetal growth restriction, child stunting, subsequent lifetime of impaired productivity, and increased maternal and fetal morbidity and mortality is continued [9].

Under nutrition among women in reproductive age is significantly higher in Africa due to chronic energy and/or micronutrient deficiencies especially during pregnancy [10, 11]. In developing nations the prevalence of under nutrition among pregnant women ranges from 13% to 38% [12, 13]. The situation is worse in Africa, where the burden of malnutrition among pregnant women is about 23% [14].

A 2018 WHO (World Health Organization) African region report indicates, nine countries in Africa had a prevalence rates above 15%, this includes Ethiopia in which maternal underweight exceeds 20% [15]. Recent study done among young pregnant mothers in Ethiopia indicates the prevalence of under nutrition is 38% [16]. Individual studies across Ethiopia indicates high rates of under nutrition among pregnant women, ranging from 9.2% to 44, 9% [17–24], making Ethiopia to be one of the countries with the highest burden of maternal under nutrition from the world.

Malnutrition is holding back development with unacceptable human consequences [1]. Globally, hunger and malnutrition reduce a Gross Domestic Product (GDP) of a given country by 1.4–2.1 trillion United States Dollar (USD) a year. Similarly, malnutrition costs between 3 and 16% of the annual GDP of the 54 African countries, and for mentioning Ethiopia loss 16.5% a year [25, 26].

Despite efforts made to improve the problem; the progress made in the last decade was very low, and currently the burden of under nutrition is continued to be the major public health problem in developing countries including our country Ethiopia [3]. Different studies done across our country tried to show the burden and determinant of under nutrition among pregnant women, in any consideration of the problems of under nutrition, these segments require special consideration. As under nutrition caused by complex interrelated factors, the programs and interventions designed to reduce its burden should depend on the reliable and recent information derived from extensive studies targeting this segment of population. Therefore, this research was aimed to assess the prevalence of under nutrition and associated factors among pregnant women attending antenatal care services at public hospitals at Western Ethiopia.

## 2. Methods and materials

### 2.1 Study design, area, and period

An institution-based cross-sectional study design was carried out. This study was conducted in public general hospitals of the Oromia region. The study was conducted in all five public hospitals found in the zone were selected as cluster sampling in the study area. This study was conducted in Public hospitals of western Ethiopia from April 10 to May 10, 2020.

### 2.2 Source and study population

The source populations were all third-trimester pregnant women who were coming for delivery and antenatal care visits in the selected public general hospitals of the Oromia region. Third-trimester pregnancy women who were coming for antenatal care visits in general public hospitals of the Oromia region western part were selected as the study population.

### 2.3 Inclusion and exclusion criteria

All selected third-trimester pregnant women who were coming for ANC in public general hospitals during the study period were included, whereas pregnancy women with bilateral edema, who were seriously sick and unable to respond to the interview, were excluded from the study.

### 2.4 Sample size and sampling techniques

Sample size was calculated using double population proportion formula for commonly associated factors of under nutrition among pregnant mothers, by assuming precision OR (Odd ratio) 1.52 (*d*) = 5%, confidence level = 95% (Z$\alpha$/2 = 1.96, Z$\beta$) = the desired power (0.84 for 80% power), and proportion of under nutrition (*P1* proportion among exposed group) 49.5%, (p2 proportion among exposed group) 39.2%. By, this double population proportion formulas it becomes 768. By considering a 5% non-response rate, the required sample size was 806 pregnant women were taken as a final sample size.

The sample size was allocated proportional to their average monthly client flow. Systematic sampling was used to select the study units from pregnant women attending ANC. The interval K value was determined for samples at each hospital by dividing the number of units in the population (N) by the desired sample size (n). The first respondent was selected by lottery method, and then every second respondent was included until the desired sample size was attained [Fig 1].

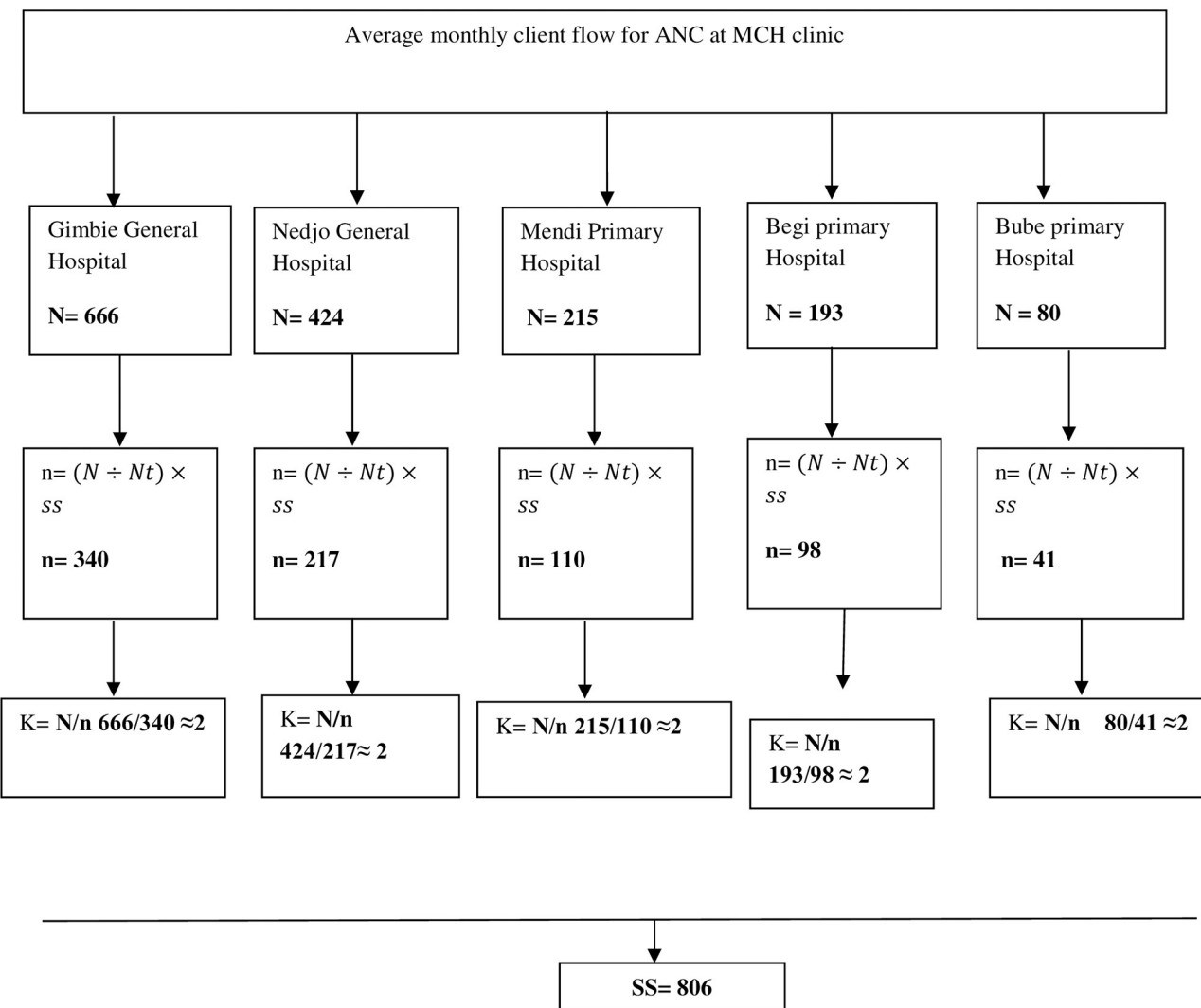

**Fig 1. Proportional allocation of sample size for assessment of under nutrition among pregnant women at public hospitals of West Wollega Zone.**

## 2.5 Data collection procedure and analysis

A semi-structured questionnaire was initially prepared in English and then translated into the regional language; Afaan Oromo was used. Afaan Oromo version was again translated back to English to check for any inconsistencies or distortion in the meaning of words. The data were collected using a pretested interviewer-administered structured questionnaire with mid-upper arm circumference measurement. Data collection was performed by one BSc Nurse as supervisor and five Midwifery nurses (Diploma) were employed for data collection. To assure the quality of the data, properly designed data collection instrument and training of data collectors and supervisors was done, the enumerators and the supervisor were given training for three days on procedures, techniques, ways of collecting the data, and monitoring the procedure especially on anthropometric measurement. Ten percent pretest was done at the, Gida public general hospital to check the consistency of the questioner. The collected data were reviewed and checked for completeness by supervisors and principal investigators each week. MUAC was measured by considering the mothers in Frankfurt plane and sideways to measure the left side, arms hanging loosely at the side with the palm facing inward, taken at marked midpoint of upper left arm, a flexible non stretchable tape were used, and difference between trainee and trainer was 0-5mm after standardization of measurement error calculation before data collection. Under nutrition was taken as a dependent variable and compared against each independent variable for association. Data were checked for completeness, consistency and accuracy. The data was entered Epi-data V.3.1., and exported to SPSS version 26 for analysis. Descriptive statistics such as mean, frequency and percentages were used to describe the study subjects. Binary and multivariable logistic regression analyses were used to see the association between the explanatory and the outcome variables. Variables with a value of $p<0.25$ during bivariate logistic regression analyses were selected as candidates variables for the multivariable logistic regression model to control for all possible confounding effects. Crude and adjusted ORs (AOR) along with 95% CIs were used to estimate the association. Variables having a value of $p<0.05$ were considered to identify factors associated with nutritional status of pregnant women.

## 2.6 Independent and dependent variable

Nutritional status of pregnant mothers is the outcome variable, and the independent variables were all the socio-demographic characteristics, dietary habit, environmental, maternal obstetrical and gynecology history. A brief description of how some of these variables were measured is as follows.

## 2.7 Dependent variable

*Nutritional status of the pregnant mothers with the measurement of* the mid-upper arm circumference values below a cutoff point < 23 cm were considered as under nutrition in this study, whereas for the individual 23 cm and above, was considered normal [13].

## 2.8 Independent variables

Potential confounding variables measured in the study were socio-demographic characteristics, obstetrics and gynecology including the age of mother, marital status, religion, educational background of mothers, women's decision-making autonomy, household income, occupation, ethnicity, number of antenatal care visits, type of pregnancy, maternal previous surgery, malaria, parity, iron and folic acid supplementation, marriage at age, substance use, coffee intake, husband's support, difficulty to access food during the last three months, Dietary

diversity, household food insecurity, prenatal feeding habits like skipping meals, frequency of meal, habit of eating snack, food avoidance, and food intake and history of low birth weight.

## 2.9 Anthropometric measurement

The anthropometric measurement mid upper arm circumstance was taken from individual third-trimester pregnant women. Intra-observer and inter-observer variability of anthropometric measurement were assessed on 10 volunteers to reduce technical error of measurement (TEM) at end of training. The measuring instruments were calibrated after each session of measurements. The Supervisor gave close supervision and technical supports, and checked the collected data for completeness, accuracy, and consistency every day and onsite.

# 3. Result

## 3.1 Socioeconomic and demographic characteristics of pregnant women

Of the 806 individuals who were approached, 780 participants were interviewed in the study giving a response rate of 96.8%. The mean (±SD) age of respondents was 26±5.32. The median family size of respondents was three persons. Majorities (64.7%) of respondents were Protestant, and about 20% were Orthodox follower. All most all of the participants (97.7% were married. Majority (53.1%) of respondents are urban dwellers while the rest (46.9%) are rural residents. About 24% of respondents completed tertiary education while only 8% had no formal education. Nearly half (48.6%) of participants were Housewife, and only 6.4% of them were daily laborers. Majority (55.4%) of respondent's family have > 37.5 $ monthly income [Table 1]. Nearly all 96.7% of participants have latrine near their house, while 83.7% have access to safe water source, and Fifty nine percent of pregnant women have low decision-making autonomy while the rest have high decision-making autonomy [Fig 2].

## 3.2 Reproductive and health care characteristics of the respondents

A mean (±SD of age at first marriage, number of pregnancy, and gestational age of respondents were 19(±2.14) years, 2(±1.16) pregnancy, and 30(±5.1) weeks respectively. About three fourth of participants were married at age of 18 years and above. Majority (63.8%) of respondents were at their third trimesters of pregnancy, about two third 68.5 of them were multigravida. Seven hundred and two of respondents (90%) said their current pregnancy was intended. About 78% of respondents were used any type of contraceptive before current pregnancy. One hundred fifty six (20%) of respondents reported history of pregnancy related complication, 8.5% reported history of current illness, and 3.8% reported history of chronic illness, while only 6.3% of them have history of substance use [Table 2].

## 3.3 Dietary characteristics of the respondents

Two hundred and thirty six (30.3%) of participants respond as consuming meals less than three times a day while majority of respondents (62.3%) of them said not increased their meals since pregnancy. Nearly half (49.2%) of pregnant women reported no habit of eating snack. Only 14.6% of participants have habit of fasting, while 18.5% have food avoidance and 8.5% have habit of skipping meal during current pregnancy.

More than three forth (80%) of pregnant women have poor prenatal feeding habit, 40% of them consumed low dietary diversity. From total participants, 600(76.9%) were from food secure, 10.8% were from mildly food insecure, 5.4% were from moderately food insecure, and

**Table 1. Socio-demographic characteristics of participants attending antenatal care services at public hospitals of West Wollega Zone, 2020.**

| Variables | | Frequency | Percent |
|---|---|---|---|
| Religion of respondents | Muslim | 72 | 9.2 |
| | Protestant | 505 | 64.7 |
| | Orthodox | 158 | 20.3 |
| | Others | 45 | 5.8 |
| Marital status of respondents | Single | 12 | 1.5 |
| | Married | 762 | 97.7 |
| | Divorced | 6 | 0.8 |
| Residence | Urban | 414 | 53.1 |
| | Rural | 366 | 46.9 |
| Respondents occupational status | Government employee | 151 | 19.4 |
| | Merchant | 118 | 15.1 |
| | House wife | 379 | 48.6 |
| | Daily laborer | 50 | 6.4 |
| | Student | 82 | 10.5 |
| Couples occupational status | Government employee | 182 | 23.7 |
| | Farmer | 253 | 32.9 |
| | Merchant | 134 | 17.4 |
| | Daily laborer | 199 | 25.9 |
| Age group of respondents | 15–24 | 330 | 42.3 |
| | 25–34 | 396 | 50.8 |
| | 35–49 | 54 | 6.9 |
| Respondents educational status | No formal education | 60 | 7.7 |
| | Primary education | 282 | 36.2 |
| | Secondary education | 252 | 32.3 |
| | Tertiary education | 186 | 23.8 |
| Couples educational status | No formal education | 54 | 7.0 |
| | Primary education | 246 | 32.0 |
| | Secondary education | 294 | 38.3 |
| | Tertiary education | 174 | 22.7 |
| Family size of respondent | ≤3 | 486 | 62.3 |
| | 4–6 | 240 | 30.8 |
| | >6 | 54 | 6.9 |
| Presence of under five children in the house hold | No | 456 | 58.5 |
| | Yes | 324 | 41.5 |
| Household monthly income | <1000 | 246 | 31.5 |
| | 1000–1500 | 102 | 13.1 |
| | >1500 | 432 | 55.4 |

Note: Others* 7[th] day Adventists

6.9% were from severely food insecure household [Table 3]. All most all (97.7%) of the participants adequately consume cereals, more than three forth (80.1%) adequately consume legumes, and more than half (58.3%) adequately consume dark green leafy vegetables. Less than half of respondents adequately consume the rest of food group listed in the table below, except milk and its products, which no participants have adequately consumed during last four weeks before the study [Table 4].

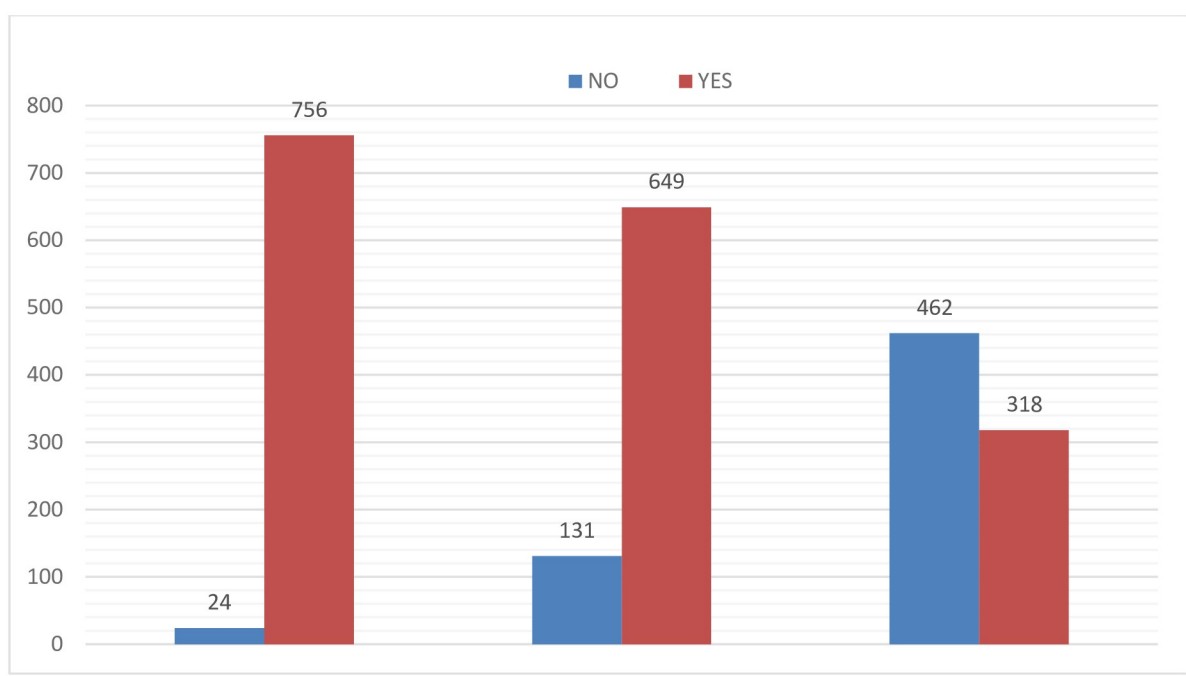

**Fig 2. Proportional allocation of sample size for assessment of under nutrition among pregnant women at public hospitals of West Wollega Zone.**

### 3.4 Nutritional status of respondents

The magnitude of under nutrition (MUAC <23cm) was 39.2%, (95%CI: 35.7%, 42.6%) [Fig 3].

### 3.5 Results of logistic regression analysis

Under nutrition was taken as a dependent variable and compared against each independent variable for association. Bivariable logistic regression was done to identify factors associated with nutritional status of pregnant women. Accordingly, household food insecurity, low dietary diversity, poor prenatal feeding habits, number of pregnancy, trimesters of pregnancy, age at first marriage less than 18 years, family size ≥6, substance use, chronic illness, rural residence, not eating snack, not increase frequency of meal shows significant association with under nutrition crudely at 25% [Tables 5–7].

Variables associated with adjusted analysis: pregnant women, accordingly, household food insecurity, low dietary diversity, substance use and residence were identified as independent predictors of under nutrition among pregnant women. The odds of under nutrition were four times [AOR = 4.36, 95%CI:(2.36, 8.79)] more among mildly food insecure household, and nearly four times [AOR = 3.71, 95%CI: 1.54, 8.61), among moderately food insecure households, and six times [AOR = 6.96, 95% CI: (3.15,15.42)] among severely food insecure household) compared with their food secure counterparts. Pregnant women with low dietary diversity had seven times [AOR = 7.56, 95% CI: (4.96, 11.51)] increased odds of under nutrition than those with high dietary diversity status.

Moreover, the odds of under nutrition was three times [AOR = 3.33, 95%CI: 1.63, 6.81)] among substance users—than their counter- parts. Rural pregnant women had nearly three

**Table 2. Reproductive and medical characteristics of participants attending antenatal care services at public hospitals of West Wollega Zone, 2020 (N = 780).**

| Variables | | Frequency | Percent |
|---|---|---|---|
| Age at first marriage of respondents (N = 774) | <18 years | 132 | 17.1 |
| | ≥ 18 years | 642 | 82.9 |
| Trimesters of pregnancy | Second | 282 | 36.2 |
| | Third | 498 | 63.8 |
| Number of pregnancy | Prim gravida | 246 | 31.5 |
| | Multigravida | 534 | 68.5 |
| Number of birth (N = 528) | Null para | 276 | 52.3 |
| | Multipara | 252 | 47.7 |
| Previous birth interval (N = 252) | <2 years | 30 | 11.9 |
| | 2 to 4 years | 150 | 59.5 |
| | > = 4 years | 72 | 28.6 |
| Intention of current pregnancy | No | 78 | 10.0 |
| | Yes | 702 | 90.0 |
| Number of antenatal care visit | First visit | 210 | 26.9 |
| | Second visit | 216 | 27.7 |
| | Third visit | 210 | 26.9 |
| | Fourth visit | 144 | 18.5 |
| Previous contraceptive use | No | 174 | 22.3 |
| | Yes | 606 | 77.7 |
| Nutritional advice during pregnancy (N = 570) | No | 276 | 48.4 |
| | Yes | 294 | 51.6 |
| Use of iron and folic acid supplementation (N = 570) | No | 36 | 6.3 |
| | Yes | 534 | 93.7 |
| Deworming (N = 570) | No | 473 | 83.0 |
| | Yes | 97 | 17.0 |
| History of pregnancy complication | No | 624 | 80.0 |
| | Yes | 156 | 20.0 |
| History of current illness | No | 714 | 91.5 |
| | Yes | 66 | 8.5 |
| History of frequent illness | No | 708 | 90.8 |
| | Yes | 72 | 9.2 |
| History of chronic illness | No | 750 | 96.2 |
| | Yes | 30 | 3.8 |
| Substance use | No | 731 | 93.7 |
| | Yes | 49 | 6.3 |

times [AOR = 2.68, 95%CI: 1.77, 4.06)] increased odds of under nutrition than urban women [Table 8].

## 4. Discussion

This study tried to reveal the prevalence and factors associated with under nutrition among pregnant women in the West Wollega Zone, the western part of Ethiopia. Accordingly, nearly forty percent (39.2%) of participants were undernourished, and factors associated with their nutritional status were residency, substance use, household food insecurity, and the low dietary diversity of women.

**Table 3. Prenatal feeding habits of participants attending antenatal care services at public hospitals of West Wollega Zone, 2020.**

| Variables | | Frequency | Percent |
|---|---|---|---|
| Frequency of meals in a day | <3 | 236 | 30.3 |
| | ≥3 | 544 | 69.7 |
| Increased frequency of meals | No | 486 | 62.3 |
| | Yes | 294 | 37.7 |
| Habit of eating snack | No | 384 | 49.2 |
| | Yes | 396 | 50.8 |
| Habit of fasting | No | 666 | 85.4 |
| | Yes | 114 | 14.6 |
| Food avoidance | No | 636 | 81.5 |
| | Yes | 144 | 18.5 |
| Habit of skipping meal | No | 714 | 91.5 |
| | Yes | 66 | 8.5 |
| Prenatal feeding habits of respondents | Poor | 630 | 80.8 |
| | Good | 150 | 19.2 |
| Dietary diversity of woman | Low | 312 | 40.0 |
| | High | 468 | 60.0 |
| Household food insecurity status | Food secure | 600 | 76.9 |
| | Mild | 84 | 10.8 |
| | Moderate | 42 | 5.4 |
| | Severe | 54 | 6.9 |

The global estimate of maternal malnutrition during pregnancy appears to be decreasing in almost all regions of the globe except in Africa, where the number of pregnant mothers with malnutrition has been increasing steadily over time. This shows that the result of this study is relevance to the current status of under nutrition in Africa, and Ethiopia in particular [14].

**Table 4. Consumption of common food groups among participants attending ANC services at public hospitals of West Wollega Zone, 2020.**

| Variable | Category | Frequency | Percent |
|---|---|---|---|
| Cereals intake | Inadequate | 18 | 2.3 |
| | Adequate | 762 | 97.7 |
| Legumes intake | Inadequate | 155 | 19.9 |
| | Adequate | 625 | 80.1 |
| Dark green leafy vegetables intake | Inadequate | 325 | 41.7 |
| | Adequate | 455 | 58.3 |
| Yellow orange vegetables intake | Inadequate | 522 | 66.9 |
| | Adequate | 258 | 33.1 |
| White roots and tubers intake | Inadequate | 532 | 68.2 |
| | Adequate | 248 | 31.8 |
| Flesh meats intake | Inadequate | 696 | 89.2 |
| | Adequate | 84 | 10.8 |
| Milk and milk products intake | Inadequate | 768 | 98.5 |
| | Adequate | 12 | 1.5 |
| Eggs intake | Inadequate | 672 | 86.2 |
| | Adequate | 108 | 13.8 |
| Oils and fats intake | Inadequate | 522 | 66.9 |
| | Adequate | 258 | 33.1 |

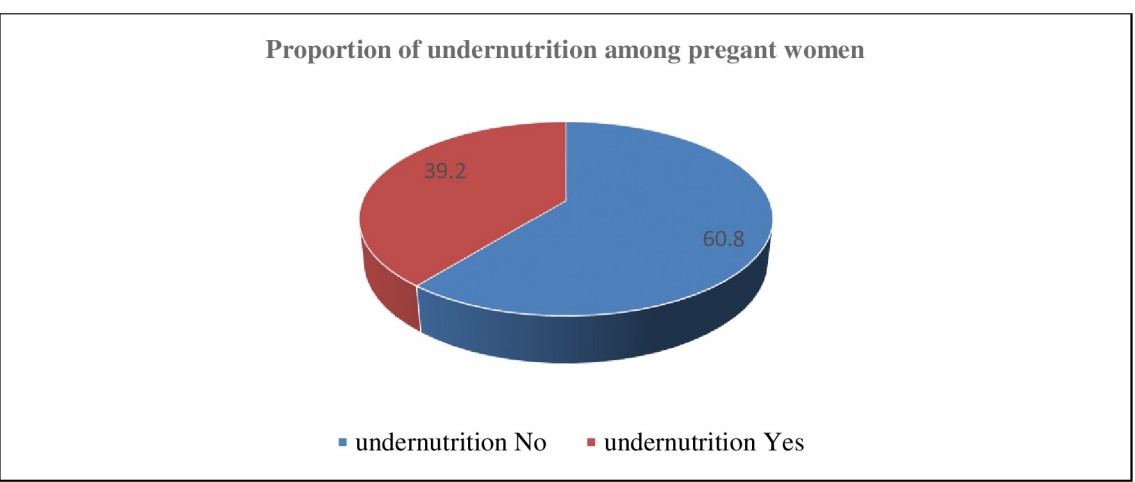

**Fig 3. Nutritional status of participants attending ANC services at public hospitals of West Wollega Zone.**

The finding of this study was almost consistent with the study conducted in the South-western part of our country, which showed 44.9% [24] and the study conducted in the Southern part of Ethiopia, which showed 35.5% [27].

On the other hand, this study result was lower than a study done in the Kunama population, Tigray, northern Ethiopia, which was observed to be 47.9%. This may be due to the fact that in the above study the proportion of food insecurity was higher than in the present study, which could increase the risk of under nutrition [28]. But a lower prevalence of under nutrition was reported among studies conducted in Gambella Town (28.6%), Alamata General Hospital (23.2%), rural communities in Haramaya district (19.06%), and Dessie Town (19.5%) [17, 18, 21, 29]. The discrepancies between the findings may be due to geographical variation between the studies or to the variation in the cut off point for MUAC measurement. The studies above used lower cut off points than the present study, which could under estimate the prevalence of under nutrition.

According to this study, rural residents were three times more likely to have under nutrition than urban ones. It is true that people's lifestyles, income, and, most importantly, health and nutrition are usually determined by where they live [30]. This finding is consistent with the findings of an African systematic review, a study at the University of Gondor Hospital, and a study in Boricha Woreda, Sidama Zone, all of which show that rural pregnant women are more likely to develop undernutrition than urban pregnant women(13,21,50). But this finding may not be true in the developed world, as one study in the USA shows there is no significant association between nutritional status and residency [31].

In this study, pregnant women who were consumed low dietary diversity were more than seven times more likely to be undernourished than those who were consumed high dietary diversity. This study's findings are consistent with a study done in Dessie Town, which found that women with low dietary diversity were nearly six times more likely to contract under nutrition than the others [17]. The study done in Gambella town also shows that pregnant women who had a low dietary diversity score were two times more at risk of under nutrition than their counterparts [23]. A survey done in Iran, and a study conducted in Kenya were also among other studies that showed similar findings to this study [32, 33]. In contrast to this study, dietary diversity did not show any significant correlation with maternal anthropometry in rural Cambodia. This may be due to the fact that a very high

**Table 5. Bivariate analysis of socio demographic factors associated with under nutrition among pregnant women attending ANC at West Wollega public hospitals, 2020.**

| Associated Factors | | Undernutrition (MUAC<23) | | P-Value | COR | 95%C.I for COR | |
|---|---|---|---|---|---|---|---|
| | | Yes Count (%) | No Count (%) | | | Lower | Upper |
| Age group of respondents | 15–24 | 138(41.8) | 192(58.2) | 0.717 | 0.89 | 0.50 | 1.60 |
| | 25–34 | 144(36.4) | 252(63.6) | 0.251 | 0.71 | 0.40 | 1.27 |
| | 35–49 | 24(44.4) | 30(55.60) | | 1 | | |
| Respondents residence | rural | 198(54.1) | 168(45.9) | 0.000 | 3.34 | 2.47 | 4.51* |
| | urban | 108(26.1) | 306(73.9) | | 1 | | |
| Household monthly income | <1000 | 112(47.5) | 124(52.5) | 0.263 | 1.29 | 0.79 | 5.25 |
| | 1000–1500 | 46(45.1) | 56(54.9) | 0.156 | 1.17 | 0.54 | 3.98 |
| | >1500 | 178(41.2) | 254(58.8) | | 1 | | |
| Family size of respondent | > = 6 | 30(55.6) | 24(44.4) | 0.039 | 1.82 | 1.03 | 3.20* |
| | 4–5 | 78(32.5) | 162(67.5) | 0.032 | 0.70 | 0.51 | 0.97 |
| | < = 3 | 198(40.7) | 288(59.3) | | 1 | | |
| Presence of under five children | Yes | 138(42.6) | 186(57.4) | 0.105 | 0.79 | 0.59 | 1.05 |
| | No | 168(36.8) | 288(63.2) | | 1 | | |
| Sources of drinking water | Unsafe | 48(80) | 12(20) | 0.479 | 1.15 | 0.78 | 1.68 |
| | Safe | 258(35.8) | 462(64.20) | | 1 | | |
| No latrine | Yes | 12(50) | 12(50) | 0.276 | 1.57 | 0.70 | 3.54 |
| | No | 294(38.9) | 462(61.1) | | 1 | | |
| Decision making autonomy | Low | 186(40.3) | 276(59.7) | 0.478 | 1.11 | 0.83 | 1.49 |
| | High | 120(37.7) | 198(62.3) | | 1 | | |
| Substance use | Yes | 32(65) | 17(34.7) | 0.000 | 3.14 | 1.71 | 5.76* |
| | No | 274(37.5) | 45762.5) | | 1 | | |
| Respondents educational status | No formal education | 66(55) | 54(45) | 0.528 | 1.21 | 0.67 | 2.20 |
| | Primary education | 162(52.9) | 144(47.1) | 0.127 | 1.35 | 0.92 | 1.97 |
| | Secondary education | 66(28.9) | 162(71.1) | 0.576 | 1.12 | 0.75 | 1.66 |
| | Tertiary education | 12(9.5) | 114(90.5) | | 1 | | |
| Couples educational status | No formal education | 72(60) | 48(40) | 0.393 | 1.31 | 0.71 | 2.43 |
| | Primary education | 108(50) | 108(50) | 0.425 | 0.85 | 0.57 | 1.27 |
| | Secondary education | 9638.1) | 156(61.9) | 0.141 | 1.33 | 0.91 | 1.96 |
| | Tertiary education | 30(16.7) | 150(83.3) | | 1 | | |
| Respondents occupational status | farmer | 66(45.8) | 78(54.2) | 0.258 | 1.33 | 0.81 | 2.17 |
| | merchant | 18(25) | 54(75) | 0.867 | 1.03 | 0.70 | 1.53 |
| | house wife | 198(51.6) | 186(48.4) | 0.651 | 1.16 | 0.60 | 2.24 |
| | daily laborer | 18(33.3) | 36(66.7) | 0.099 | 1.58 | 0.92 | 2.73 |
| | government employee | 6(6.7) | 84(93.3) | | 1 | | |
| Couples occupational status | farmer | 192(56.1) | 150(43.9) | 0.094 | 1.40 | 0.94 | 2.06 |
| | merchant | 18(15) | 102(85) | 0.551 | 1.15 | 0.73 | 1.82 |
| | daily laborer | 72(52.2) | 66(47.8) | 0.774 | 1.06 | 0.70 | 1.61 |
| | government employee | 24(14.8) | 138(85.2) | | 1 | | |

Note:

* = statistically significant at p-value < 0.25,

1: reference category, COR: Crude odds ratio, 95%CI: 95 percent confidence interval

**Table 6. Bivariate analysis of reproductive and medical factors associated with under nutrition among participants attending ANC services at West Wollega public hospitals, 2020.**

| Associated Factors | | Undernutrition (MUAC<23) | | P-Value | COR | 95% CI for COR | |
|---|---|---|---|---|---|---|---|
| | | Yes, Count (%) | No. Count (%) | | | Lower | Upper |
| Number of pregnancy | multigravida | 228(42.7) | 306(57.3) | 0.004 | 1.61 | 1.17 | 2.21* |
| | prim gravida | 78(31.7) | 168(68.3) | | 1 | | |
| Trimesters of pregnancy | 3rd trimester | 174(34.9) | 324(65.1) | 0.001 | 0.61 | 0.45 | 0.82* |
| | 2nd trimester | 132(46.8) | 150(53.2) | | 1 | | |
| Age at first marriage | < 18 years | 96(72.7) | 36(27.3) | 0.000 | 5.49 | 3.62 | 8.32* |
| | > = 18 years | 210(32.7) | 432(67.3) | | 1 | | |
| History of illness in current pregnancy | Yes | 30(45.5) | 36(54.5) | 0.280 | 1.32 | 0.80 | 2.20 |
| | No | 276(38.7) | 438(61.3) | | 1 | | |
| History of chronic illness | Yes | 18(60) | 12(40) | 0.021 | 2.41 | 1.14 | 5.06* |
| | No | 288(38.4) | 262(61.6) | | 1 | | |
| History of pregnancy complication | Yes | 60(38.5) | 96(61.5) | 0.826 | 0.96 | 0.67 | 1.38 |
| | No | 246(39.40) | 378(60.60) | | 1 | | |
| No previous contraceptive use | Yes | 72(41.40) | 102(58.60) | 0.510 | 1.12 | 0.80 | 1.58 |
| | No | 234(38.60) | 372(61.40) | | 1 | | |
| Number of antenatal care visit | first visit | 84(40) | 126(60) | 0.276 | 0.79 | 0.51 | 1.21 |
| | second visit | 66(30.60) | 150(69.4) | 0.003 | 0.52 | 0.34 | 0.81 |
| | third visit | 90(42.9) | 120(57.1) | 0.580 | 0.89 | 0.58 | 1.36 |
| | fourth visit | 66(45.8) | 78(54.2) | | 1 | | |
| Non intended pregnancy | Yes | 36(46.2) | 42(53,8) | 0.188 | 1.37 | 0.86 | 2.20 |
| | No | 270(38.5) | 432(61.5) | | 1 | | |

Note:

* = statistically significant at p-value < 0.25,

1: reference category, COR: Crude odds ratio, 95%CI: 95 percent confidence interval

proportion of women show prevalence of low dietary diversity in the study done in rural Cambodia [34].

In this study, respondents who were from food insecure household shows more at risk of malnutrition than those from food secure households. Our finding of an increased prevalence of maternal under nutrition in food insecure households may reflect inequitable intra-household food allocation whereby the nutritional needs of the child and/or other members of the household are prioritized over those of the mother. As evidenced by the2019 FAO food insecurity reported that, household food insecurity was found to be associated with more than one form of malnutrition [30]. Similar findings have been reported from studies conducted in the Gumay district, Gambella Town, and the Kunama population in the Tigray region [23, 24, 28]. A study done in Nepal also indicates a significant relationship between food adequacy and low nutritional status of pregnant women [35]. In contrast to this, increased maternal anthropometry was observed among women from mildly food insecure households in the USA, Brazil and Lebanon. This discrepancy may be due to the fact that these studies were from the developed world with different sociodemographic characteristics, and they have different method of assessment when compared to present study [36–38].

The educational status of respondents does not show a significant association with under nutrition in current study. However, this finding contradicts the findings of a study conducted in the Shashamene district of southern Ethiopia, which found that literate women had a 70% lower risk of under nutrition than those with no formal education [39]. Another study done at

**Table 7. Bivariate analysis of dietary factors with nutritional status of pregnant women attending antenatal care clinics in public hospitals of West Wollega Zone, 2020.**

| Associated Factors | | Undernutrition (MUAC<23) | | P-Value | COR | 95% CI for COR | |
|---|---|---|---|---|---|---|---|
| | | Yes, Count (%) | No, Count (%) | | | Lower | Upper |
| Less than three meals in a day | Yes | 95(40.3) | 141(59.7) | 0.700 | 1.06 | 0.78 | 1.45 |
| | No | 211(38.8) | 333(61.2) | | 1 | | |
| No habit of eating snack | Yes | 186(48.4) | 198(51.6) | 0.000 | 2.16 | 1.61 | 2.89* |
| | No | 120(30.3) | 276(69.7) | | 1 | | |
| Not increased frequency of meals | Yes | 240(49.4) | 246(50.6) | 0.000 | 3.37 | 2.43 | 4.67* |
| | No | 66(22.4) | 228(77.6) | | 1 | | |
| Food avoidance during pregnancy | Yes | 54(37.5) | 90(62.5) | 0.638 | 0.91 | 0.63 | 1.33 |
| | No | 252(39.60) | 384(60.4) | | 1 | | |
| Habit of fasting while pregnant | Yes | 54(47.4) | 60(52.6) | 0.055 | 1.48 | 0.99 | 2.20 |
| | No | 252(37.8) | 414(62.2) | | 1 | | |
| Habit of skipping meal | Yes | 54(81.8) | 12(18.2) | 0.000 | 8.25 | 4.33 | 15.71* |
| | No | 252(35.3) | 462(64.7) | | 1 | | |
| Prenatal feeding habits | Poor | 187(29.7) | 443(70.3) | 0.236 | 0.84 | 0.25 | 1.67 |
| | Good | 50(33.3) | 150(66.7) | | 1 | | |
| Household food insecurity status | Severe | 42(77.9) | 12(22.2) | 0.000 | 8.57 | 4.41 | 16.67* |
| | Moderate | 30(71.4) | 12(28.8) | 0.000 | 6.12 | 3.06 | 12.23* |
| | Mild | 60(71.4) | 24(28.6) | 0.000 | 6.12 | 3.69 | 10.14* |
| | Food secure | 174(29) | 426(71) | | 1 | | |
| Dietary diversity of woman | Low | 222(71.2) | 90(28.8) | 0.000 | 11.28 | 8.03 | 15.84* |
| | High | 84(17.9) | 384(82.0) | | 1 | | |
| Dark green leafy vegetables intake | inadequate | 121(37.2) | 204(62.8) | 0.334 | 0.87 | 0.65 | 1.16 |
| | adequate | 185(40.7) | 270(59.3) | | 1 | | |
| Yellow orange vegetables intake | inadequate | 216(41.4) | 306(58.6) | 0.081 | 1.32 | 0.97 | 1.80 |
| | adequate | 90(34.9) | 168(65.1) | | 1.0 | | |
| White roots and tubers intake | inadequate | 207(38.9) | 325(61.1) | 0.788 | 0.96 | 0.70 | 1.31 |
| | adequate | 99(39.9) | 149(60.1) | | 1 | | |
| Flesh meats intake | inadequate | 270(38.8) | 426(61.2) | 0.472 | 0.85 | 0.53 | 1.34 |
| | adequate | 36(42.9) | 48(57.1) | | 1 | | |
| Eggs intake | inadequate | 265(39.4) | 407(60.6) | 0.771 | 1.06 | 0.70 | 1.62 |
| | adequate | 41(38) | 67(62.0) | | 1 | | |
| Oils and fats intake | inadequate | 202(38.7) | 320(61.3) | 0.664 | 0.94 | 0.69 | 1.27 |
| | adequate | 104(40.3) | 154(59.7) | | 1 | | |

Note:

* = statistically significant at p-value < 0.25,

1: reference category, COR: Crude odds ratio, 95%CI: 95 percent confidence interval

the University of Gondor Hospital also shows the risk of under nutrition was nearly three times higher among pregnant women with no formal education [22]. The study done in rural Nepal also reported a significant relationship between the educational level of the women and their general nutritional status [34]. The reason might be that food and related factors that have an effect on the nutritional status of women are under the control of the household head, even though the women have higher education.

**Table 8. Multivariate analysis of factors associated with under nutrition among pregnant women attending ANC services at public hospitals of West Wollega Zone.**

| Associated factors | | Undernutrition (MUAC<23cm) | | Bivariate analysis | | | Multivariable analysis | | |
|---|---|---|---|---|---|---|---|---|---|
| | | Yes | No | COR | 95% C.I COR | | AOR | 95% C.I AOR | |
| | | | | | Lower | Upper | | Lower | Upper |
| Household food insecurity status | Severe | 42(77.9%) | 12(22.2%) | 8.57 | 4.41 | 16.67 | 6.96 | 3.15 | 15.42** |
| | Moderate | 30(71.4%) | 12(28.8%) | 6.12 | 3.06 | 12.23 | 3.71 | 1.54 | 8.96** |
| | Mildly | 60(71.4%) | 24(28.6%) | 6.12 | 3.69 | 10.14 | 4.55 | 2.36 | 8.79** |
| | Food secure | 174(29%) | 426(71%) | 1 | | | 1 | | |
| Dietary diversity of woman | Low | 222(71.2%) | 90(28.8%) | 11.28 | 8.03 | 15.85 | 7.56 | 4.96 | 11.51** |
| | High | 84(17.9%) | (384)82.0% | 1 | | | 1 | | |
| Prenatal feeding habits | Poor | 343(54.4%) | 287(45.6%) | 5.77 | 3.48 | 9.57 | 1.81 | 0.86 | 3.83 |
| | Good | 131(87.3%) | 19(12.7%) | 1 | | | 1 | | |
| Number of pregnancy | Multigravida | 228(42.7%) | 306(57.3%) | 1.61 | 1.17 | 2.21 | 0.98 | 0.60 | 1.60 |
| | Prim gravida | 78(31.7%) | 168(68.3%) | 1 | | | 1 | | |
| Trimesters of pregnancy | Third trimester | 174(34.9%) | 324(65.1%) | 0.61 | 0.45 | 0.82 | 0.78 | 0.52 | 1.17 |
| | Second trimester | 132(46.8%) | 150(53.2%) | 1 | | | 1 | | |
| Age at first marriage | < 18 years | 96(72.7%) | 36(27.3%) | 5.49 | 3.62 | 8.32 | 1.62 | 0.93 | 2.82 |
| | > = 18 years | 210(32.7%) | 432(67.3%) | 1 | | | 1 | | |
| Family size | > = 6 | 30(55.6%) | 24(44.4%) | 1.82 | 1.03 | 3.2 | 0.47 | 0.22 | 1.00 |
| | 4_5 | 78(32.5%) | 162(67.5%) | 0.7 | 0.51 | 0.97 | 0.77 | 0.48 | 1.23 |
| | < = 3 | 198(40.7%) | 288(59.3%) | 1 | | | 1 | | |
| Substance use | Yes | 32(65%) | 17(34.7%) | 3.14 | 1.71 | 5.76 | 3.33 | 1.62 | 6.81** |
| | No | 274(37.5%) | 457(62.5%) | 1 | | | 1 | | |
| History of chronic illness | Yes | 18(60%) | 12(40%) | 2.41 | 1.14 | 5.07 | 2.83 | 0.98 | 8.12 |
| | No | 288(38.4%) | 262(61.6%) | 1 | | | 1 | | |
| Respondents residence | Rural | 198(54.1%) | 168(45.9%) | 3.34 | 2.47 | 4.51 | 2.68 | 1.77 | 4.06** |
| | Urban | 108(26.1%) | 306(73.9%) | 1 | | | 1 | | |
| No habit of eating snack | Yes | 186(48.4%) | 198(51.6%) | 2.16 | 1.61 | 2.9 | 1.07 | 0.67 | 1.69 |
| | No | 120(30.3%) | 276(69.7%) | 1 | | | 1 | | |
| Not increased frequency of meals | Yes | 240(49.4%) | 246(50.6%) | 3.37 | 2.43 | 4.67 | 1.21 | 0.70 | 2.07 |
| | No | 66(22.4%) | 228(77.6%) | 1 | | | 1 | | |

Note:

** indicates statistically significant at P-value < 0.05,

1 indicates reference, COR = crude odd ratio, AOR = adjusted odd ratio

## 5. Limitations of the study

- Even though this study tried to cover several variables, it does not include variables that need laboratory investigations such as intestinal parasites, and malaria infection.

- Dietary intake of respondents was measured only on occasional time, and that may not show the seasonal variability on availability of food.

- This study used anthropometric measure to assess nutritional status of pregnant mothers and the effect of technical error was not ruled out that may affect the reliability of result.

- As it is institutional based study, the finding of this study does not fully indicate the characteristics of respondents at community level which means it is not generalized beyond study population.

## 6. Conclusions and recommendations

### 6.1 Conclusion

This study revealed that, the prevalence of under nutrition assessed by mid-upper arm circumference was nearly forty percent among pregnant women which is showing high prevalence of under nutrition among target population that needs priority attention for intervention. The factors that significantly associated with under nutrition were respondent's residency, substance use, low dietary diversity, and household food insecurity status. The remaining factors studied did not show any significant association with under nutrition.

### 6.2 Recommendations

Based on the finding of this study the following recommendations were forwarded.

- Governments should adopt coherent policies, which foster cross-sectoral cooperation and strategies to avert the problem of under nutrition among pregnant women.

- Policy makers and implementers should make programs that improve food security status at household level to increase access to high nutritious food and variety of foods among poor.

- Essential investments must be made into nutrition-sensitive programme in other areas such as agriculture, education, water and social protection.

- Agricultural sector should implement nutrition sensitive interventions such as increased productivity and dietary diversifications.

- Health workers should disseminate useful information about the harmful effect of using substance during pregnancy at health institution and community level in order to improve the dietary habits of pregnant women especially for rural residents.

- Health workers should also advice pregnant women about the benefits of dietary diversity during pregnancy.

## Supporting information

**S1 Annex. Study questionnaire in English.**
(DOCX)

**S1 File.**
(SAV)

**S1 Data.**
(XLSX)

## Acknowledgments

We are grateful to officials from Zonal health office and public Hospitals of West Wollega zone, and the health workers of each facility for their valuable contribution during the study. We also extend our thanks to data collectors, respondents, and supervisors for their cooperation during the study.

## Ethics approval and consent to participate

The study protocol was approved and permission was obtained from Wollega University. An official letter of co-operation was written to the selected Public Hospitals of West Wollega

Zone Administration. Information on the studies, including purpose and procedures was given for participants. Written or verbal consent was obtained from each participant. In order to protect confidentiality, names or identifications were not included on the written questionnaires. Identification of the respondents was only through numerical codes.

## Author Contributions

**Conceptualization:** Lamessa Tafara.

**Data curation:** Lamessa Tafara.

**Formal analysis:** Lamessa Tafara, Haile Bikila, Ilili Feyisa, Markos Desalegn, Zalalem Kaba.

**Investigation:** Lamessa Tafara, Haile Bikila, Ilili Feyisa, Markos Desalegn, Zalalem Kaba.

**Methodology:** Lamessa Tafara, Haile Bikila, Ilili Feyisa, Markos Desalegn, Zalalem Kaba.

**Software:** Lamessa Tafara, Haile Bikila, Ilili Feyisa, Markos Desalegn, Zalalem Kaba.

**Supervision:** Lamessa Tafara, Haile Bikila, Ilili Feyisa, Markos Desalegn, Zalalem Kaba.

**Validation:** Lamessa Tafara, Haile Bikila, Ilili Feyisa, Markos Desalegn, Zalalem Kaba.

**Visualization:** Lamessa Tafara, Haile Bikila, Ilili Feyisa, Markos Desalegn, Zalalem Kaba.

**Writing – original draft:** Lamessa Tafara, Haile Bikila, Ilili Feyisa, Markos Desalegn, Zalalem Kaba.

**Writing – review & editing:** Lamessa Tafara, Haile Bikila, Ilili Feyisa, Markos Desalegn, Zalalem Kaba.

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
