## [Decision Letter · Decision Letter 0]

3 Feb 2022

PONE-D-21-23542The magnitude of under nutrition and associated factors among pregnant women attending Antenatal Care service in Public Hospitals of western Ethiopia.PLOS ONE

Dear Dr. Tafara,

Thank you for submitting your manuscript to PLOS ONE. After careful consideration, we feel that it has merit but does not fully meet PLOS ONE’s publication criteria as it currently stands. Therefore, we invite you to submit a revised version of the manuscript that addresses the points raised during the review process.

The manuscript has been evaluated by two reviewers, and their comments are available below.

The reviewers have raised a number of major issues in relation to the reporting of the study methodology. In particular, the study inclusion and exclusion criteria requires additional details in order to provide further clarification on the sampled cohort. Furthermore please provide details related to the questionnaire used in the study and describe whether the tool used has been previously validated.

Could you please revise the manuscript to carefully address the concerns raised?

We look forward to receiving your revised manuscript.

Kind regards,

Lucinda Shen

Staff  Editor

PLOS ONE

Journal Requirements:

3. You indicated that you had ethical approval for your study. In your Methods section, please ensure you have also stated whether you obtained consent from parents or guardians of the minors included in the study or whether the research ethics committee or IRB specifically waived the need for their consent.

6. Please amend your list of authors on the manuscript to ensure that each author is linked to an affiliation. Authors’ affiliations should reflect the institution where the work was done (if authors moved subsequently, you can also list the new affiliation stating “current affiliation:….” as necessary).’

7. Your ethics statement should only appear in the Methods section of your manuscript. If your ethics statement is written in any section besides the Methods, please move it to the Methods section and delete it from any other section. Please ensure that your ethics statement is included in your manuscript, as the ethics statement entered into the online submission form will not be published alongside your manuscript. 

Reviewers' comments:

Reviewer's Responses to Questions

**Comments to the Author**

1. Is the manuscript technically sound, and do the data support the conclusions?

Reviewer #1: Partly

Reviewer #2: Partly

2. Has the statistical analysis been performed appropriately and rigorously? 

Reviewer #1: Yes

Reviewer #2: I Don't Know

3. Have the authors made all data underlying the findings in their manuscript fully available?

Reviewer #1: Yes

Reviewer #2: No

4. Is the manuscript presented in an intelligible fashion and written in standard English?

Reviewer #1: Yes

Reviewer #2: No

5. Review Comments to the Author

Reviewer #1: The theme is relevant, however follow some suggestions.

Abstract:

The objective is described twice in the abstract

Key words: replace “under nutrition”, as it is already in the title.

Background: I

suggest finishing with the objective of the study.

Methods and materials:

Describe the location hospitals of the Oromia region.

Contextualizing about antenatal care visits, is always performed in hospitals, low and medium risk pregnant women receive the same standard of care, ...

It was not clear whether the pregnant women included were the ones who would have the delivery or only those who attended the antenatal care visits.

Acronym: ANC,MUAC, BSc

In the exclusion criteria explain better what they consider seriously sick.

Were high-risk pregnant women, for example with diabetes and/or hypertension included?

How was the classification of under nutrition performed only by the MUAC?

Were no other parameters used, such as the Growth Charts and Gestational Age?

I suggest justifying and referencing the use of MUAC more intensely, since it is not an indicator used for pregnant women and is not robust for under nutrition .

What does substance use mean?

What are dietary habits evaluated and classified in what way? Which reference authors? To assess household food insecurity there are instruments widely used in the literature.

Consumption of common food groups. Which authors used to classify it as adequate or inadequate?

Result: Excluiria a figura 3

In the methodology they mention hemoglobin level, but I did not find this information in the results

Discussion

line 258-259: The prevalence of malnutrition described in other studies used the same parameters, i.e., MUAC for pregnant women?

Conclusion:

...“nearly forty percent of pregnant women were undernourished”... – would add by MUAC indicator

Reviewer #2: The manuscript presents an important issue in maternal and child health and nutrition. The high prevalence of undernutrition among pregnant women in Ethiopia is alarming and the investigation of associated factors may support the local public policies. However, I missed some important information in the methods and results, which I believe weakens the quality of reporting. So please, find attached my suggestions for the manuscript, just with the intention of improving the paper.

6. PLOS authors have the option to publish the peer review history of their article (what does this mean?). If published, this will include your full peer review and any attached files.

Reviewer #1: No

Reviewer #2: No

---

## [Author Response · Author response to Decision Letter 0]

16 Sep 2022

Point to point Responses to Reviewers

Reviewer #1: 

Abstract: The objective is described twice in the abstract

Response: Corrected as The objective which is written at the end of the abstract was the recommendation and its corrected and replaced by intended to be written by. 

Reviewer #1: Key words: replace “under nutrition”, as it is already in the title.

Response: Corrected 

Reviewer #1: Describe the location hospitals of the Oromia region.

Response: It’s found at west wollega zone, Oromia regional state Ethiopia 

Reviewer #1: Contextualizing about antenatal care visits, is always performed in hospitals, low and medium risk pregnant women receive the same standard of care, ...

Response: There is no way that the service could be provided at different levels because the study area focused on institutionalized ANC mothers who met the inclusion and exclusion criteria. If there is a problem, it is already dealt with under the exclusion criteria. Unless otherwise stated, there is no specialized ANC service based on their health status; nevertheless, if the mother is at danger, referral or consulting for additional management may be appointed. 

Reviewer #1: It was not clear whether the pregnant women included were the ones who would have the delivery or only those who attended the antenatal care visits.

Response: It did not consider the post natal or at delivery time, the source populations are only mothers attending ANC service.

Reviewer #1: Acronym: ANC, MUAC, BSc

Response: Thank you Corrected 

Reviewer #1: In the exclusion criteria explain better what they consider seriously sick.

 Response: This means that if the mother is unable to talk or answer due to her health condition at the time of data collection. 

Reviewer #1: Were high-risk pregnant women, for example with diabetes and/or hypertension included?

Response: Yes, unless other ways they did not faced any acute problem which exclude thmen from the study unit like convulsion and fen tines. 

Reviewer #1: How was the classification of under nutrition performed only by the MUAC?

Response: Different finding shows that MUAC is best for pregnant mother than BMI to assess their nutritional status. During recent times, MUAC has been used for evaluation of adult nutritional status as well, especially in resource-limited settings, including India. Prior studies also suggest that MUAC can be an efficient indicator of adult under nutrition comparable or even better than BMI. The results suggest that it is possible to conduct community-level screening of malnourishment among adult/adolescent women using less resource-intensive techniques such as MUAC. 

Reviewer #1: I suggest justifying and referencing the use of MUAC more intensely, since it is not an indicator used for pregnant women and is not robust for under nutrition. 

Response: Body-mass-index (BMI) is widely accepted as an indicator of nutritional status in adults. Mid-upper-arm-circumference (MUAC) is another anthropometric-measure used primarily among children. The present study attempted to evaluate the use of MUAC as a simpler alternative to BMI cut-off <18.5 to detect adult under nutrition, and thus to suggest a suitable cut-off value.

There a study which is conducted to detedt the sensitivity and specificity using, Curve estimation was done to assess the linearity and correlation of BMI and MUAC. Sensitivity and specificity of MUAC against BMI<18.5 was determined. Separate Receiver-operating-characteristic (ROC) analyses were performed for male and female. Area under ROC curve and Youden's index were generated to aid selection of the most suitable cut-off value of MUAC for under nutrition.

However, BMI has some drawbacks and practical limitations as a measurement tool in the quick assessment of individuals (e.g. debilitated, disabled or acutely ill patients). It is not always possible to measure weight or height, particularly in debilitated and immobile patients. The reason is nearly always that patients cannot be taken out of their beds to be weighed and/or cannot stand for height measurements. BMI is particularly inappropriate for pregnant women. Due to the extra weight of the fetus, other products of conception, and added maternal tissue, Furthermore, in resource limited health settings and population-based surveys, accurate measurements of height and weight require reasonably large logistical mobilization.

Reviewer #1: What does substance use mean?

Response: The use of like alcohol, cigarette and other related substance 

Reviewer 1: What are dietary habits evaluated and classified in what way? Which reference authors? To assess household food insecurity there are instruments widely used in the literature.

Response: For dietary habit Food Frequency Questionnaire (FFQ) tool were applied and the House Hold food security is assessed by FANTA protocol. 

Reviewer #1: Consumption of common food groups. Which authors used to classify it as adequate or inadequate?

Response: This adequacy and inadequacy were assessed based 10 items of Dietary diversity recommended for pregnant woman by FAO , and those mothers consumes less the five(5) were considered as inadequate and greater than five(5) were considered as adequate

Reviewer #1: Line 258-259: The prevalence of malnutrition described in other studies used the same parameters, i.e., MUAC for pregnant women? 

Response: Most of them Yes, However there is the scientific finding and approach in which we can use both MUAC and BMI in parallel in limited resource country and its better for pregnant mother for the reason well explained under response number 10. 

Reviewer #1. How was this variable evaluated in "poor" or "good"?, How was this variable evaluated in "low" or "high"? Adequate inadequate, etc 

Response: This assessed by standard tool developed like food frequency questionnaire (FFQ), house hold food security is assessed by FANTA procedure which has nine indicators. Similarly adequacy and inadequacy of the food is measured based on list of 10 food types developed by FAO, and those who consume less than 5 food type classified as low dietary diversity or inadequacy and vice versa. 

Reviewer #1: Conclusion: 

Response: Thank you, Corrected 

Reviewer #2: I missed some important information in the methods and results, which I believe weakens the quality of reporting. So please, find attached my suggestions for the manuscript, just with the intention of improving the paper.

Response: In the method part I added more information as follow, The data were collected using a pretested interviewer-administered structured questionnaire. Data collection was performed by one BSc Nurse as supervisor and five Midwifery nurses (Diploma) were employed for data collection. These who were familiar with the study area and could speak the local language ‘Afan Oromo’which is official language widely spoken in the area. In this study, the adequacy and inadequacy of minimum Diet Diversity Score was defined as the number of different food groups consumed over a given reference period. It was created by summing up the number of food groups consumed over 24 hours (a day before data collection) by the mother to 10 and dichotomizing according to the Food and Agricultural Organization (FAO) and Family Health International (FHI) 360, 2016 guidelines.[41], thus, consumption of at least five or more food groups indicates adequate dietary diversity.as well as the house hold food security and food habit were assessed by FANTA Food frequency questionnaire respectively. 

 Under nutrition was taken as a dependent variable and compared against each independent variable for association. Data were checked for completeness, consistency and accuracy. The data was entered Epi-data V.3.1., and exported to SPSS version 26 for analysis. Descriptive statistics such as mean, frequency and percentages were used to describe the study subjects. Binary and multivariable logistic regression analyses were used to see the association between the explanatory and the outcome variables. Variables with a value of p<0.25 during bivariate logistic regression analyses were selected as candidates variable for the multivariable logistic regression model to control for all possible confounding effects. Crude and adjusted ORs (AOR) along with 95% CIs were used to estimate the association. A variables having a value of p<0.05 was considered to identify factors associated with nutritional status of pregnant women. 

In the result part I have changed the figure 2 into a bar chart which more elaborate the finding and its more self-explanatory than the before figure.

---

## [Decision Letter · Decision Letter 1]

14 Nov 2022

The prevalence of under nutrition and associated factors among pregnant women attending Antenatal Care service in Public Hospitals of western Ethiopia.

PONE-D-21-23542R1

Dear Dr. Tafara,

We’re pleased to inform you that your manuscript has been judged scientifically suitable for publication and will be formally accepted for publication once it meets all outstanding technical requirements.

Kind regards,

Claudio Romero Farias Marinho, Ph.D.

Academic Editor

PLOS ONE

Additional Editor Comments (optional):

Reviewers' comments:

Reviewer's Responses to Questions

**Comments to the Author**

1. If the authors have adequately addressed your comments raised in a previous round of review and you feel that this manuscript is now acceptable for publication, you may indicate that here to bypass the “Comments to the Author” section, enter your conflict of interest statement in the “Confidential to Editor” section, and submit your "Accept" recommendation.

Reviewer #2: All comments have been addressed

2. Is the manuscript technically sound, and do the data support the conclusions?

Reviewer #2: Yes

3. Has the statistical analysis been performed appropriately and rigorously? 

Reviewer #2: Yes

4. Have the authors made all data underlying the findings in their manuscript fully available?

Reviewer #2: Yes

5. Is the manuscript presented in an intelligible fashion and written in standard English?

Reviewer #2: No

6. Review Comments to the Author

Reviewer #2: Dear authors, congratulations for your great job with the Ethiopian pregnant women. I suggest an English review of the manuscript before publication.

7. PLOS authors have the option to publish the peer review history of their article (what does this mean?). If published, this will include your full peer review and any attached files.

Reviewer #2: No

---

## [Editor Report · Acceptance letter]

5 Jan 2023

PONE-D-21-23542R1 

The Prevalence of under nutrition and associated factors among pregnant women attending Antenatal Care service in Public Hospitals of western Ethiopia 

Dear Dr. Tafara:

I'm pleased to inform you that your manuscript has been deemed suitable for publication in PLOS ONE. Congratulations! Your manuscript is now with our production department. 

Kind regards, 

on behalf of

Dr. Claudio Romero Farias Marinho 

Academic Editor

PLOS ONE